# Insights into the Interaction Mechanisms of the Proviral Integration Site of Moloney Murine Leukemia Virus (Pim) Kinases with Pan-Pim Inhibitors PIM447 and AZD1208: A Molecular Dynamics Simulation and MM/GBSA Calculation Study

**DOI:** 10.3390/ijms20215410

**Published:** 2019-10-30

**Authors:** Qingqing Chen, Yan Wang, Shanshan Shi, Kaihang Li, Ling Zhang, Jian Gao

**Affiliations:** Jiangsu Key Laboratory of New Drug Research and Clinical Pharmacy, Xuzhou Medical University, Xuzhou 221004, China; cq1561552131@163.com (Q.C.); wy08080148@163.com (Y.W.); s767403579@163.com (S.S.); a18192367391@163.com (K.L.); zhamgling1999@163.com (L.Z.)

**Keywords:** Pim kinase, PIM447, AZD1208, molecular dynamics simulation, MM/GBSA calculation

## Abstract

Based on the up-regulation of the proviral integration site of the Moloney murine leukemia virus (Pim) kinase family (Pim1, 2, and 3) observed in several types of leukemias and lymphomas, the development of pan-Pim inhibitors is an attractive therapeutic strategy. While only PIM447 and AZD1208 have entered the clinical stages. To elucidate the interaction mechanisms of three Pim kinases with PIM447 and AZD1208, six Pim/ligand systems were studied by homology modeling, molecular docking, molecular dynamics (MD) simulation and molecular mechanics/generalized Born surface area (MM/GBSA) binding free energy calculation. The residues of the top group (Leu44, Val52, Ala65, Lys67, and Leu120 in Pim1) dominated the pan-Pim inhibitors binding to Pim kinases. The residues of the bottom group (Gln127, Asp128, and Leu174 in Pim1) were crucial for Pims/PIM447 systems, while the contributions of these residues were decreased sharply for Pims/AZD1208 systems. It is likely that the more potent pan-Pim inhibitors should be bound strongly to the top and bottom groups. The residues of the left, right and loop groups were located in the loop regions of the binding pocket, however, the flexibility of these regions triggered the protein interacting with diverse pan-Pim inhibitors efficiently. We hope this work can provide valuable information for the design of novel pan-Pim inhibitors in the future.

## 1. Introduction

The proviral integration site of the Moloney murine leukemia virus (Pim) oncogene family plays an essential role in tumor growth and development which consists of three constitutively active serine/threonine kinases (Pim1, 2, and 3) [1]. The *PIM1* gene has been found to function as a proto-oncogene. When it is overexpressed, it induces lymphomas in transgenic mice, though at a low frequency. However, when Pim1 is co-expressed in mice with the oncogene c-Myc, 100% of the transgenic mice die of lymphomas in utero [2]. Interestingly, the knockout of Pim1 in mice is not lethal nor does its absence induce any immediately obvious phenotype. The absence of Pim1 may be compensated for by Pim2, although not in all cases. In one case where the Pim2 kinase did compensate, it appeared to contribute to cell survival, indicating that in some instances it functions similarly to Pim1 [2]. Pim3 is also one of candidate compensatory kinases for Pim1. When Pim1 and Pim3 are knocked out, mice are viable healthy and fertile, which indicates that Pim2 alone can compensate for most, but not all, of the activities of Pim1 and Pim3. They have differing but mutually compensatory functions across tumors. Thus, pan-Pim inhibitors are superior to selective Pim inhibitors in tumor inhibition.

Several different classes of Pim kinase inhibitors have recently been reported, including ruthenium-containing organometallic complexes [3], bisindolylmaleimides [4], imidazo[1,2–*b*]pyridazines [5], benzoisoxazoles [6], and cinnamic acids [7]. Pan-Pim kinase inhibitors have been extensively reported, including GDC-0339 [8], GNE-955 [9], AZD1208 [10], and PIM447 [11], among these pan-Pim inhibitors only PIM447 and AZD1208 successfully entered the clinical stage.

Compared with the AZD1208, PIM447 is a more potent pan-Pim kinase inhibitor, showing better pharmacokinetic activity. The molecular mechanism of their activity differences has not been reported. Besides, the dynamics bindings of PIM447 and AZD1208 to Pim kinases (Pim1, 2, and 3) are still unknown. In the present work, six Pims/ligands systems including Pim1/AZD1208, Pim1/PIM447, Pim2/AZD1208, Pim2/PIM447, Pim3/AZD1208 and Pim3/PIM447 were constructed and further studied using molecular dynamics (MD) simulation and molecular mechanics/generalized Born surface area (MM/GBSA) calculation. We expect this work to pave novel ways for better understanding of the binding interactions between Pim kinases and pan-Pim inhibitors, and would benefit the structure-based design of novel pan-Pim inhibitors in the near future.

## 2. Results and Discussion

### 2.1. Molecular Modeling of Pim3

The target sequence was aligned to the template of Pim3 using the align2d script in Modeller 9.20 (Figure 1A) [12]. The sequence similarity between Pim1 and 3 was very high (78%). Based on the alignment, five resultant models were built (Table 1 and Figure 1B). The third model with the genetic algorithm 341 (GA341) score of 1.00000, the Modeller objective function (molpdf) score of 1775.07397, and the discrete optimized protein energy (DOPE) score of −33461.44922 was selected. Following this, the FG-MD [13] (https://zhanglab.ccmb.med.umich.edu/FG-MD/) method was used to optimize the Pim3.B99990003 model (Figure 1C).

From the protein structure analysis (ProSA) plot (Figure 1D), the Z-score of Pim3.B99990003 model was −7.18, indicating a high quality model was obtained. A Ramachandran plot (Figure 1E) validated the Pim3.B99990003 model with 80.7% of the total residues in the most favored region, 17.2% in the additional allowed region and 1.5% in the generously allowed region (Leu21, Arg169, Asp297, and Val300). Only two residues (Ser35 and Thr310) were located in the disallowed region, which constituted 0.7% of the total protein, which represented a good quality of the predicted model. Moreover, the Pim3.B99990003 model showed 92.9825% of quality factor using error atom type (ERRAT) (Figure 1F). Thus, the Pim3.B99990003 model was used for the following molecular docking and MD simulations.

### 2.2. Overall Structure and Dynamics

To explore the dynamic stability of these six systems, the root mean square deviation (RMSD) values of the whole protein backbone atoms were calculated (Figure 2A). The RMSD plot indicated that all six systems achieved equilibrium in a short time. The RMSD values of all systems except for the Pim3/PIM447 system fluctuated around 3 Å. The lower RMSD values implied that the binding patterns of Pim/ligand complexes derived from the molecular docking studies changed little during 20 ns MD simulations. To further assess the stability of six systems after 10 ns, the same RMSD calculations were conducted on the whole protein backbone atoms with respect to the structure at t = 10 ns (Figure 2B). It was clear that all six systems already equilibrated after 10 ns and fluctuated around 1.5 Å, which implied that 20 ns MD simulations were enough to relax the Pim/ligand complexes. It is noteworthy that the RMSD value of Pim2/AZD1208 was more or less larger than other systems, which may be attributed to the bigger fluctuations of residues Phe49 and Val52 obtained by a root mean square fluctuation (RMSF) calculation (Appendix A). Therefore, it was reasonable to perform the following binding free energy calculation and free energy decomposition analysis based on the last 10 ns trajectories.

### 2.3. Binding Free Energies Estimated by MM/GBSA

The calculated MM/GBSA binding free energies (ΔG_bind_) of Pim1/AZD1208, Pim1/PIM447, Pim2/AZD1208, Pim2/PIM447, Pim3/AZD1208, and Pim3/PIM447) were −31.17 ± 2.64, −37.15 ± 2.46, −25.19 ± 2.67, −35.92 ± 2.88, −28.61 ± 2.95, and −38.58 ± 2.19 kcal/mol, respectively (Table 2). The correlation between the calculated binding free energies and the experimental pKi (i.e., −logKi) was high (R^2^ = 0.795). Meanwhile, the MM/GBSA calculation also performed well in investigating the binding affinities of three Pim kinases to GDC-0039 and hispidulin (Appendix A). Table 2 also showed that the van der Waals interactions played a dominant role not only in the total binding free energies of the six systems, but in ranking the origin of difference in the binding affinities of the systems. The net electrostatic free energy (ΔE_ele_ + ΔG_GB_) of each system had a negative contribution on the binding of PIM447 (or AZD1208) to Pim kinases. It may be attributed to the fact electrostatic interaction is fully counteracted by the solvation effect. It is noteworthy that the differences in entropy contribution (TΔS) of the six systems were not obvious, which implied the normal-mode analysis might be not accurate enough for Pim kinases. The molecular mechanisms of the Pim kinases lower inhibitory activity of AZD1208 compared to PIM447, and the better binding affinities of AZD1208 to Pim1 than to the other two kinases (Pim2 and 3), were illuminated in following the free energy decomposition and binding mode analysis.

### 2.4. Decomposition Analysis of the Binding Free Energies

MM/GBSA free energy decomposition analysis was employed to decompose the total binding free energies into ligand–residue pairs, which would provide more detailed information about the contribution of each residue of Pim kinases on the interacting with inhibitors. To investigate the direct interactions in the Pim/ligand complexes, 14 amino acid residues which contributed significantly (<−1 kcal/mol) to the binding of inhibitors to Pim kinases were selected (Figure 3). Considering that the 14 amino acid residues could form the binding pocket of the inhibitor, these residues were naturally divided into five groups (Figure 4) according to the binding pattern of PIM447 in Pim1, i.e., top group (Leu44, Val52, Ala65, Lys67, Leu120), left group (Arg122, Pro123, Val126), bottom group (Gln127, Asp128, Leu174), right group (Ile185, Asp186), and loop group (Phe49).

Alignment of amino acid sequences of the Pim kinases (Pim 1, 2, and 3) showed that among the 14 residues only Val126 (numbered in Pim1) in the left group was different (Figure 5). The residue valine of Pim1 was replaced by alanine in Pim2 and 3, which resulted in the decreased contribution of the residue (Figure 3). However, the total contributions of the left group on six Pim/ligand systems were not significantly different (Figure 6, Table 3). In other words, the weaker contribution of the residue alanine in Pim2 and 3 was nearly complemented by the other two residues (Arg and Pro) in the left group (Figure 3 and Figure 7). The difference of residue at the position 126 (numbered in Pim1) had little effect on the binding interactions between Pim kinases and pan-Pim inhibitors.

In order to further analyze the reasons for the different activity of Pim inhibitors, the differences in contribution values of the five groups were studied (Figure 6, Table 3). As shown in Figure 6 and Figure 7, the top group had the largest contribution in all of the six systems, and the top, right, and left groups had a greater contribution to the sum binding free energies both of Pims/PIM447 and Pims/AZD1208. It was obviously that the contribution of these three groups in the six Pims/ligands systems were similar, while the bottom and loop groups contributed differently to the sum binding free energies of Pims/PIM447 and Pims/AZD1208. The contribution value of each group in the three Pims/PIM447 was in the same order. The order from high to low was top, bottom, right, left, and loop group (Figure 6, Table 3). However, the contribution of AZD1208 in Pim1 and Pim3 were ranked as top > right > Left > loop > bottom group. The order of contribution value of five groups in Pim2/AZD1208 system was top > right > left > bottom > loop group. It was notable to conclude that the difference between the contribution values of the bottom and loop groups was the main factor affecting the activity (Table 3). The total contribution of the bottom group in each Pim/PIM447 system was about three times larger than that of the corresponding Pim/AZD1208 system (Table 3), which indicated that the bottom group was crucial for improving the activity of pan inhibitors. Although the loop group had only one amino acid residue in each system, the contribution of it varied greatly in the six systems. The maximal contribution value of the loop group was −2.47 kcal/mol in the Pim3/AZD1208 system and the minimal contribution value was −0.05 kcal/mol in the Pim2/AZD1208 system, which meant that the residue in the loop group had a significant influence on the activity of inhibitors.

## 3. Materials and Methods 

### 3.1. Homology Model of Pim3

Considering that the crystal structure of Pim3 was not available, a homology model of Pim3 was carried out firstly. The amino acid sequence of th ePim3 protein was retrieved from UniProt (http://www.uniprot.org) with the UniProt entry of Q86V86. For template selection, the build_profile script was performed by searching against the pdball database using Modeller 9.20 [14]. Results from the template search showed Pim1 was the best template for construction of the 3D structure of Pim3 due to its high amino acid sequence identity (78%) to Pim3. The 3D structure of the template protein Pim1 can be obtained from the Protein Data Bank (PDB) (www.rcsb.org/pdb) with the PDB ID of 3UIX. 

Aligning of amino acid sequences and construction of homology models were conducted by align2d script [15] and model-single script with default values in Modeller 9.20, respectively. The best model can be selected by picking the model with the highest value of the GA341 score (the highest score was 1.00000) and the lowest values of the molpdf and DOPE scores [16,17,18].

### 3.2. Optimization of Homology Model of Pim3 

The newly built homology models often produce unfavorable atomic distances, bond angles, van der Waals radius overlap, and undesirable torsion angles. Therefore, it was essential to minimize the energy to regularize local bond and angle geometry as well as to relax close contacts in the geometric chain. The FG-MD (https://zhanglab.ccmb.med.umich.edu/FG-MD/) method [13] was employed to optimize the generated structure. Structural quality of the optimized protein was analyzed by ProSA web. Various other parameters including buried protein, quality factor, three-dimensional score, and non-bonded interaction were analyzed by SAVES (https://servicesn.mbi.ucla.edu/SAVES/).

### 3.3. Construction of Six Pim/Ligand Systems

The Surflex-Dock module of Sybyl X2.1 software (SYBYL_X2.1 is available from Tripos Associates Inc., S. H. R.; St. Louis, MO 631444, USA.) was used for molecular docking studies to predict the binding modes of AZD1208 and PIM447 to Pim kinases. The parameters of docking were set as default. Herein, six systems, i.e., Pim1/AZD1208, Pim1/PIM447, Pim2/AZD1208, Pim2/PIM447, Pim3/AZD1208, and Pim3/PIM447 were constructed. The crystal structure of Pim1/AZD1208 could not be obtained directly, while the ligand (7li) in the crystal structure of 4DTK was similar to AZD1208. Based on the binding pose of ligand 7li, the Pim1/AZD1208 system was constructed via the sketch module in Sybyl X2.1. The crystal structure of Pim1/PIM447 could be obtained directly (PDB ID: 5DWR). The crystal structure of Pim2 which was used for molecular docking was 4X7Q. The 3D structure of Pim3 was generated from the above homology modeling study. Then, AZD1208 and PIM447 were docked to Pim2 and 3, respectively.

### 3.4. Molecular Dynamics Simulations

MD simulations were performed on the above six systems using the Amber12 software package [19]. The tleap module was used to added missing hydrogen atoms of the six complexes. The proteins and the ligands were respectively parameterized using the standard amber force field (ff03) and general amber force field (gaff). Each ligand was minimized using the HF/6–31 * optimization in the Gaussian 09 program [20] and the atomic charges were subsequently fitted by using the restrained electrostatic potential (RESP). Then, each system was immersed into a cubic TIP3P water box extended 12 Å from any solute atom, and appropriate amounts of Na^+^ were added to neutralize the system. Prior to MD simulations, the systems were optimized by the conjugate gradient and steepest descent methods as described in previous papers [21,22]. With the integration step time was given as 2 fs and the constant temperature was run at 310 K, 20 ns MD simulations were performed. The entire coordinate file was saved at each 1 ps for the following binding free energy calculation and free energy decomposition analysis.

### 3.5. MM/GBSA Calculation and MM/GBSA Free Energy Decomposition Analysis

For the purpose of elucidating the binding affinities of the six Pim/ligand systems, the MM/GBSA binding free energy calculation was conducted, and the method was calculated according to the following equation [23]:
ΔG_bind_ = G_complex_ − G_protein_ − G_ligand_= ΔE_MM_ + ΔG_GB_ + ΔG_SA_ − TΔS= ΔE_vdw_ + ΔE_ele_ + ΔG_GB_ + ΔG_SA_ − TΔS(1)
where ΔE_MM_ is the gas-phase interaction energy that is composed of van der Waals (ΔE_vdw_) and electrostatic (ΔE_ele_) energies; ΔG_GB_ is polar desolvation free energy that is calculated by the generalized Born (GB) model [24], and ΔG_SA_ is the non-polar desolvation free energy which is estimated from the accessible surface area (SASA) model by the LCPO method: ΔG_SA_ = 0.0072 × ΔSASA [25]. −TΔS is the conformational entropy contribution at temperature T. In the GB model, the solvent and the solute dielectric constants were set to 80 and 4, respectively. The binding free energy of each system was calculated based on 1000 snapshots evenly extracted from 10 to 20 ns MD trajectories. In view of the high computational demand of normal-mode analysis, only 100 snapshots extracted from the last 10 ns were used to estimate the entropy contribution. To validate the feasibility of the MM/GBSA methods for Pim/ligand systems, the MD simulation and MM/GBSA calculation studies were also applied to the other two Pim1 inhibitors GDC-0039 [8] and hispidulin [26], the potencies of which are lower than the compounds AZD1208 and PIM447. The detailed results about MD simulations and free energy calculations are described in the Appendix A.

To uncover the key residues in the binding interactions between Pim kinases and inhibitors, MM/GBSA free energy decomposition analysis was carried out using the mm_pbsa module in Amber12. After the decomposition process, the total binding free energy contribution could be allocated to each residue from the association of the receptor and the ligand. All 400 snapshots generated for the binding free energy calculation were also used for the free energy decomposition analysis.

## 4. Conclusions

We studied the molecular recognitions of two pan-Pim inhibitors (PIM447 and AZD1208) with three Pim kinases via homology modeling, molecular docking, MD simulation and MM/GBSA binding free energy calculation. Fourteen amino acids which bound strongly to the inhibitor binding pocket were divided into five groups (top, left, bottom, right, and loop). The top group had the greatest contribution on the binding of pan-Pim inhibitors to Pim kinases. The bottom group contributed greater to Pims/PIM447 than to Pims/AZD1208, which gave rise to the higher activity of PIM447. The contribution of the loop group varied greatly in the six systems, which suggested it had an important impact on the activity of different inhibitors. The amino acids of the left group and the right group were located in the loop regions of the proteins, however, the flexibility of these regions gave the protein an advantage to accommodate diverse pan-Pim inhibitors. Not surprisingly, the contributions of the left and right group were similar both in Pims/PIM447 and Pims/AZD1208. It is likely that the more potent pan-Pim inhibitors should be bound strongly to the top and bottom groups. Our work could be helpful in clarifying the distinguishing roles of the 14 amino acids in improving the activity of inhibitors and designing novel inhibitors, which will be valuable for Pim-driven human cancers.

## Figures and Tables

**Figure 1 ijms-20-05410-f001:**
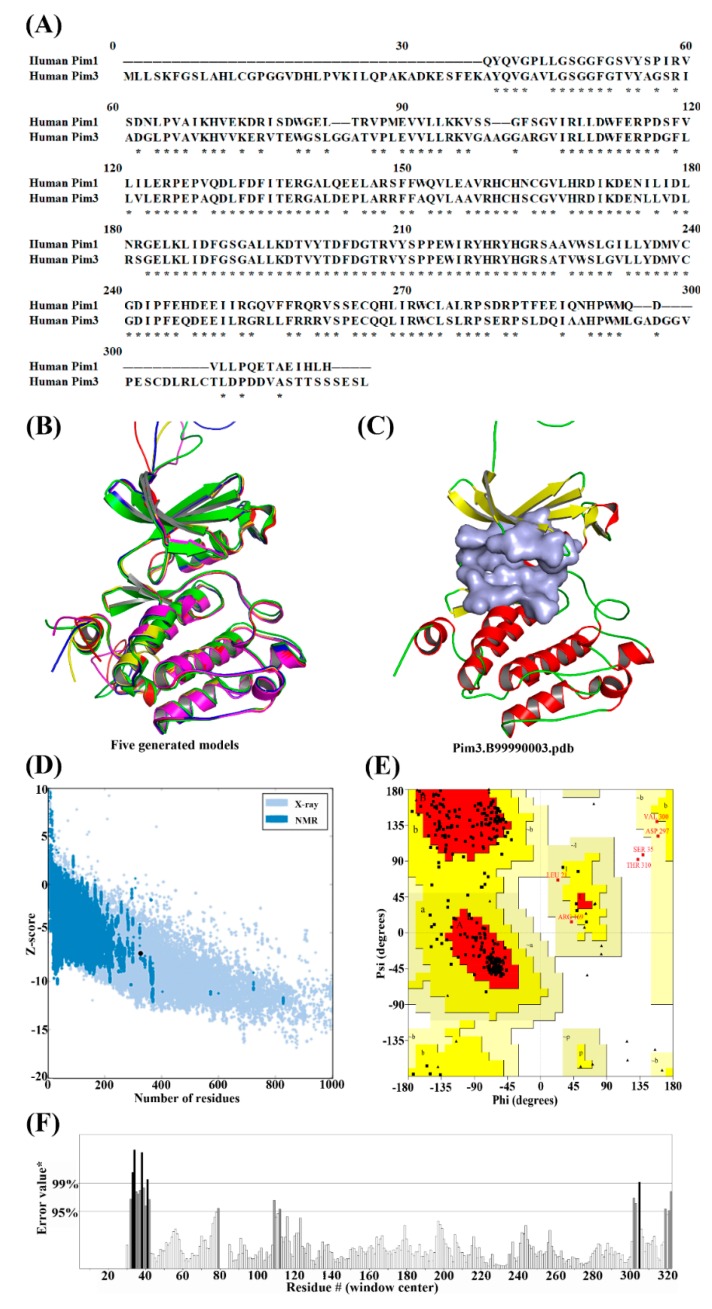
(**A**) The sequence alignments of human Pim3 and Pim1 (PDB ID: 3UIX). The same amino acids are indicated by an asterisk. (**B**) Five resultant Pim3 models generated by Modeller 9.20. (**C**) The Pim3.B99990003 model, the best model selected by picking the model with the highest GA341 score and the lowest value of the molpdf and the DOPE scores. (**D**) The protein structure analysis (ProSA) plot of the generated Pim3 model, the Z-score value indicated the quality of the model. (**E**) The Ramachandran plot of the generated Pim3 model. (**F**) Quality verification plot of the generated Pim3 model using error atom type (ERRAT).

**Figure 2 ijms-20-05410-f002:**
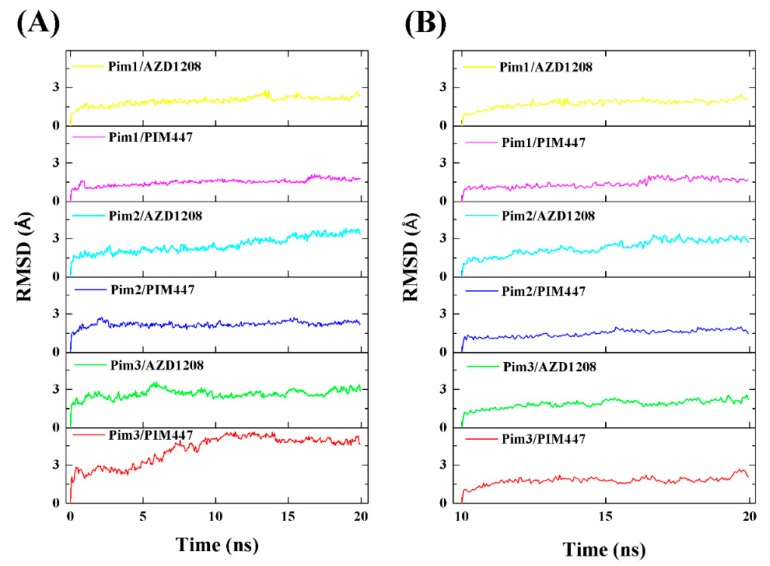
(**A**) Time evolution of the backbone root mean square deviation (RMSD) of the six systems with respect to the starting structure. (**B**) The same RMSD values in the molecular dynamic (MD) trajectories of the six systems from 10 to 20 ns.

**Figure 3 ijms-20-05410-f003:**
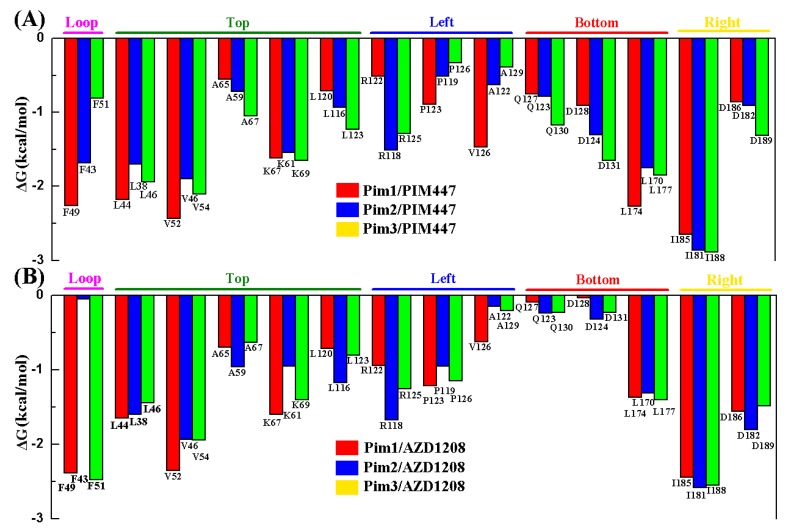
MM/GBSA decomposition results of the total binding free energies per residue for Pims/PIM447 (**A**) and Pims/AZD1208 (**B)** systems.

**Figure 4 ijms-20-05410-f004:**
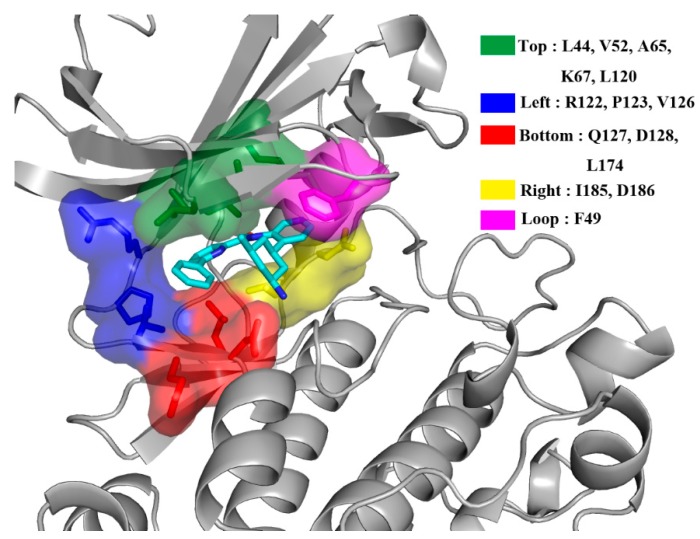
Contribution of the binding free energy of each residue was further classified into five groups (top, left, bottom, right and loop group) according to the binding pocket.

**Figure 5 ijms-20-05410-f005:**
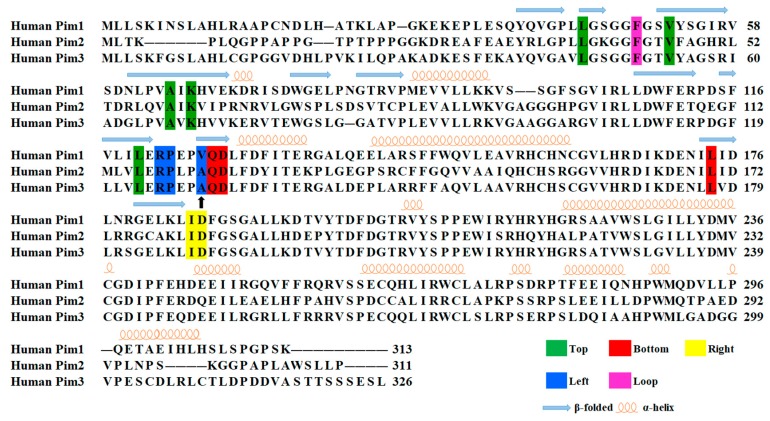
First-order sequence alignment of three Pim kinases. Fourteen key residues calculated by MM/GBSA free energy decomposition were divided into five groups: top (green), left (blue), bottom (red), right (yellow), and loop group (pink). The blue arrows indicate beta strands and orange spirals indicate alpha helices. The black arrow indicate that the residues at this site differ among the three Pim kinases.

**Figure 6 ijms-20-05410-f006:**
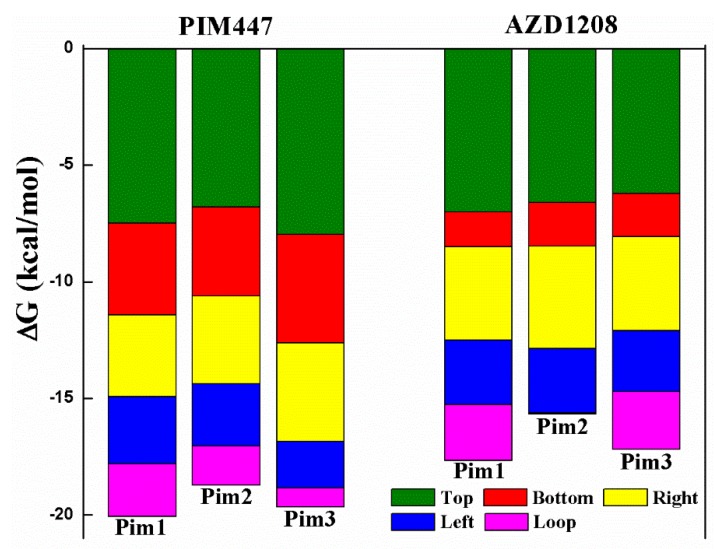
The proportion of binding free energy for each group in six Pim/ligand systems.

**Figure 7 ijms-20-05410-f007:**
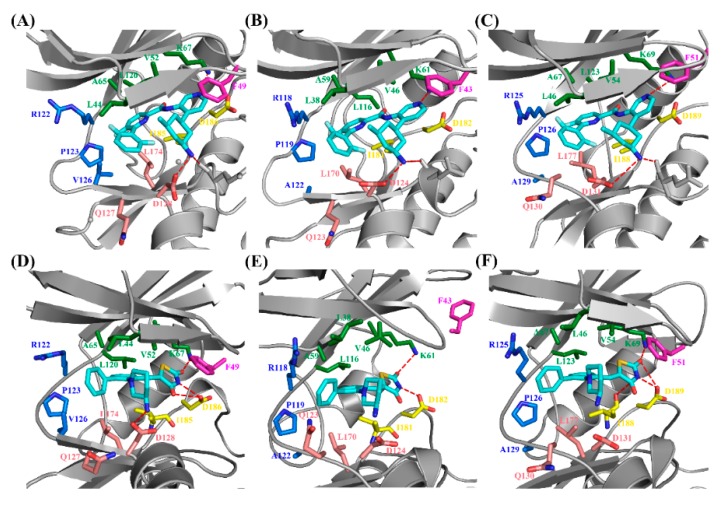
Illustration of the binding interactions between three Pim kinases and pan-Pim inhibitors in the Pim1/PIM447 (**A**), Pim2/PIM447 (**B**), Pim3/PIM447 (**C**), Pim1/AZD1208 (**D**), Pim2/AZD1208 (**E**), and Pim3/AZD1208 (**F**) systems. The proteins Pim1, Pim2, and Pim3 are shown in cartoon modes and are colored in grey, while the key residues for the inhibitor (cyan) binding are shown in stick mode and are colored in forest (top group), blue (left group), red (bottom group), yellow (right group), and magenta (loop group).

**Table 1 ijms-20-05410-t001:** The Modeller objective function (molpdf), discrete optimized protein energy (DOPE) score and genetic algorithm 341 (GA341) score of five proviral integration site of the Moloney murine leukemia virus (Pim)3 models.

Models	molpdf ^a^	DOPE Score ^b^	GA341 Score ^c^
Pim3.B99990001	1709.42212	−33,078.14063	1.00000
Pim3.B99990002	1679.96997	−33,142.85938	1.00000
Pim3.B99990003	1775.07397	−33,461.44922	1.00000
Pim3.B99990004	1816.04749	−33,243.78516	1.00000
Pim3.B99990005	1728.24146	−33,196.65234	1.00000

^a^ molpdf: MODELLER minimizes the objective function with respect to Cartesian coordinates of ~10,000 atoms (3D points) that form a system. ^b^ DOPE score: assesses the quality of the selected atoms in the model using the discrete optimized protein energy (DOPE) method and is a statistical potential optimized for model assessment. ^c^ GA341 score: assesses the quality of the model using the GA341 method.

**Table 2 ijms-20-05410-t002:** Binding free energy of components of six Pim/ligand systems calculated in molecular mechanics/generalized Born surface area (MM/GBSA) (kcal/mol).

System	ΔE_vdw_	ΔE_ele_	ΔG_GB_	ΔG_SA_	TΔS	ΔG_bind_	K_i_ (pM)
Pim1/AZD1208	−39.65 ± 3.16	−5.45 ± 3.33	9.39 ± 1.99	−5.90 ± 0.18	−10.44 ± 2.28	−31.17 ± 2.64	17
Pim1/PIM447	−47.98 ± 2.68	−9.99 ± 2.56	15.94 ± 2.16	−6.36 ± 0.17	−11.24 ± 2.71	−37.15 ± 2.46	6
Pim2/AZD1208	−37.27 ± 2.98	−6.16 ± 3.02	10.83 ± 2.38	−5.92 ± 0.19	−13.33 ± 2.14	−25.19 ± 2.67	160
Pim2/PIM447	−46.81 ± 3.00	−10.69 ± 2.27	15.86 ± 1.78	−6.38 ± 0.12	−12.10 ± 2.45	−35.92 ± 2.88	18
Pim3/AZD1208	−39.24 ± 3.21	−9.55 ± 2.50	12.61 ± 1.87	−5.90 ± 0.13	−13.47 ± 2.63	−28.61 ± 2.95	230
Pim3/PIM447	−49.55 ± 2.33	−9.41 ± 1.59	15.76 ± 1.17	−6.63 ± 0.11	−11.26 ± 2.36	−38.58 ± 2.19	9

**Table 3 ijms-20-05410-t003:** Contributions of binding free energies for five groups calculated by MM/GBSA free energy decomposition (kcal/mol).

System	Top	Bottom	Right	Left	Loop	Sum
Pim1/PIM447	−7.49	−3.93	−3.51	−2.87	−2.26	−20.06
Pim2/PIM447	−6.79	−3.83	−3.77	−2.65	−1.68	−18.72
Pim3/PIM447	−7.97	−4.67	−4.20	−2.01	−0.81	−19.66
Pim1/AZD1208	−7.01	−1.50	−4.00	−2.77	−2.38	−17.66
Pim2/AZD1208	−6.61	−1.87	−4.38	−2.77	−0.05	−15.68
Pim3/AZD1208	−6.21	−1.86	−4.03	−2.61	−2.47	−17.18

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
