# Peer review of "Insights into the Interaction Mechanisms of the Proviral Integration Site of Moloney Murine Leukemia Virus (Pim) Kinases with Pan-Pim Inhibitors PIM447 and AZD1208: A Molecular Dynamics Simulation and MM/GBSA Calculation Study"

_ijms, 2019, doi:10.3390/ijms20215410_

Round 1

Reviewer 1 Report

This paper reports a molecular dynamics simulation of Pim 1, 2, and 3 proteins with two pan-Pim inhibitors, PIM447 abd AZD1208. These two inhibitors are in the clinical stage, but, surprisingly, their binding patterns to Pim kinases have not been reported. Although this is a pure in silico study, I believe that this paper presents novel and useful findings in this field. Manuscript is written in acceptable English, although minor language proofreading will be required before considering publication.

Pim3 model: The authors selected the Pim3.B99990003 model based on the GA341 and DOPE scores. This is the best model among all models tested in this study, but I am not sure whether this model is robust enough to reach the conclusion. I encourage authors to cite other studies to justify their strategy. Especially, as the authors stated in L195, the crystal structure of Pim3 is not available and amino acid identity between Pim1 and Pim3 (78%) is not very high.

L98: Binding appears to reach the equilibrium after 10 ns in most case as the authors stated, but the Pim2/AZD data seems to have an increasing tendency even at 20 ns. Can you provide a more detailed rationale for selecting the last 10 ns trajectories?

Author Response

Jian Gao

Jiangsu Key Laboratory of New Drug Research and Clinical Pharmacy, Xuzhou Medical University, Xuzhou, Jiangsu 221004, P. R. China

Oct 21, 2019

Dear editor:

We would like to thank you and reviewers for the critical comments and thoughtful suggestions. According to the comments and suggestions, we have made careful modifications on the revision.

Enclosed please find: 1) Our response to the comments point by point;

Reviewer #1: Comments:

Pim3 model: The authors selected the Pim3.B99990003 model based on the GA341 and DOPE scores. This is the best model among all models tested in this study, but I am not sure whether this model is robust enough to reach the conclusion. I encourage authors to cite other studies to justify their strategy. Especially, as the authors stated in L195, the crystal structure of Pim3 is not available and amino acid identity between Pim1 and Pim3 (78%) is not very high.

Response: We would like to thank you to point out. We had cited several previous papers in which the GA341 and DOPE scores were used to evaluate the obtained models. We had cited three literatures in the section of “3.1 Homology model of Pim3” as follow: The best model can be selected by picking the model with the highest value of the GA341 score (the highest score was 1.00000) and the lowest values of the molpdf and DOPE scores [16-18].  

The amino acid identity between Pim1 and Pim3 (78%) is not very high, while the binding sites of inhibitor in the two enzymes is very high (92.86%). In detail, fourteen residues were regarded as the key residues for inhibitor binding which calculated by MM/GBSA free energy decomposition (Figure 3), and only the Val126 (numbered in Pim1) is different between Pim1 and 3 (Figure 5). It may be acceptable to model the Pim3 structure based on the Pim1.

L98: Binding appears to reach the equilibrium after 10 ns in most case as the authors stated, but the Pim2/AZD data seems to have an increasing tendency even at 20 ns. Can you provide a more detailed rationale for selecting the last 10 ns trajectories?

Response: We would like to thank you to point out. The RMSD value of Pim2/AZD1208 system was more or less larger than those of other systems as shown in Figure 2A. To further assess the stability of six systems after 10 ns, the same RMSD calculation were conducted on the whole protein backbone atoms with respect to the structure at t =10 ns (Figure 2B). The RMSD value of Pim2/AZD1208 system calculated by the last 10 ns fluctuated below 3 Å, and maintained smooth in the last 5 ns. Meanwhile, the root mean square fluctuation (RMSF) of per amino acid residue was also calculated based on the last 10 ns trajectories to evaluate the binding stability of six systems. It can be seen that the RMSF values of the six systems were similar, and the RMSF values of the residues around the active site region are lower than those of in other regions. However, the residues Phe49 and Val52 (numbered in Pim1) in Pim2/AZD1208 system had big RMSF values (about 4.5 Å), which may result in the larger RMSD fluctuation of Pim2/AZD1208 system. As for the Pim2/PIM447, only Phe49 was unstable. In fact, the Phe49 (numbered in Pim1) belonged to the Loop group and would be flexible in nature. The instability of Phe49 and Val52 (numbered in Pim1) triggered the lowest binding free energy of Pim2/AZD1208 system (-25.19 ± 2.67, from Table 2), while the error bar of energy is smaller enough (2.67) when compared to other systems. Thus, the last 10 ns trajectories may be acceptable to do the binding free energy calculation.

       Meanwhile, we appended several sentences in the section of “2.2 Overall structure and dynamics” as follow: “It is noteworthy that the RMSD value of Pim2/AZD1208 was more or less larger than other systems, which may be attributed to the bigger fluctuations of residues Phe49 and Val52 obtained by root mean square fluctuation (RMSF) calculation (Figure S1).”

Figure S1 RMSF of the backbone atoms of protein versus the residue numbers of Pims in six systems. The purple dotted lines indicate the key residues for inhibitor binding.

Thank you very much for reconsideration.

Sincerely yours,

Jian Gao

Reviewer 2 Report

The Authors have made an attempt to study the interaction mechanisms of pan PIM inhibitors PIM447 and AZD1208 using molecular modelling tools.

Comments:

Authors have selected two PIM inhibitors which have entered to clinical trials and attempted to find the interaction mechanisms. These two compounds are having an IC50 value between pM to nM (PIM447 reported with IC50 between 6 to 18pM whereas, AZC1208 is reported between 0.4 to 1.9nM). These two compound are very potent. When authors tried to validate their molecular dynamic simulation, it is better to include some of the low potent PIM inhbitors like GDC-0339 (IC50 - 43.6nM) and Hispidulin (IC50-2.71µM). So when a method is validated it is better to cover low to high potency compounds (between pM to micro Mol potency). Present study indicated both the compounds have almost similar interaction character (Fig.6 proportion of binding energy is almost similar except bottom). Apart from the study design, other part is very well planned, performed and presented. It is also a good idea to mention the limitations associated with the simulation methods , so that, other researchers will interpret the results/outcome with caution

Author Response

Jian Gao

Jiangsu Key Laboratory of New Drug Research and Clinical Pharmacy, Xuzhou Medical University, Xuzhou, Jiangsu 221004, P. R. China

Oct 21, 2019

Dear editor:

We would like to thank you and reviewers for the critical comments and thoughtful suggestions. According to the comments and suggestions, we have made careful modifications on the revision.

Enclosed please find: 1) Our response to the comments point by point;

Reviewer #2: Comments:

Authors have selected two PIM inhibitors which have entered to clinical trials and attempted to find the interaction mechanisms. These two compounds are having an IC50 value between pM to nM (PIM447 reported with IC50 between 6 to 18pM whereas, AZC1208 is reported between 0.4 to 1.9nM). These two compound are very potent. When authors tried to validate their molecular dynamic simulation, it is better to include some of the low potent PIM inhbitors like GDC-0339 (IC50 - 43.6nM) and Hispidulin (IC50-2.71µM). So when a method is validated it is better to cover low to high potency compounds (between pM to micro Mol potency).

Response: We would like to thank you to point out that it is optimal to validate the MM/GBSA binding free energy method by evaluation different Pim inhibitor (inhibitory activity varies from pM to micro Mol). In our revised manuscript, we studied the binding affinities of compounds GDC-0339 and Hispidulin to three Pim enzymes, and MM/GBSA binding free energies were shown in Table S1. It is likely that the calculated binding free energies had high correlation with the experimental pKi (i.e., -lgKi). MM/GBSA binding free energy method also performed well in ranking the pim inhibitors with different inhibitory activities. Thus, we appended one sentence in the section of “2.3 Binding free energies estimated by MM/GBSA” as follow: Meanwhile, the MM/GBSA calculation also performed well in investigating the binding affinities of three Pim kinases to GDC-0039 and Hispidulin (Table S1). We also appended two sentences in the section of “3.5 MM/GBSA calculation and MM/GBSA free energy decomposition analysis” as follow: To validate the feasibility of the MM/GBSA methods for Pim/ligand systems, the MD simulation and MM/GBSA calculation studies were also applied to the other two Pim1 inhibitors GDC-0039 [26] and Hispidulin [27], the potencies of which are lower than the compounds AZD1208 and PIM447. The detailed results about MD simulations and free energy calculations were described in the Supplementary data.

Table S1. Binding free energies of six Pim/ligand systems calculated in MM/GBSA (kcal/mol).

System

ΔEvdw

ΔEele

ΔGGB

ΔGSA

TΔS

ΔGbind

Ki (nM)

Pim1/GDC-0339

-42.2±2.71

-15.33±3.78

19.85±3.45

-6.55±0.22

-12.56±2.28

-31.51±2.86

0.03

Pim1/Hispidulin

-34.45±2.91

-12.38±5.31

18.68±4.50

-4.87±0.19

-12.24± 2.71

-20.77±2.73

2710

Pim2/GDC-0339

-35.76±2.91

-18.66±3.19

22.65±2.73

-6.84±0.17

-10.14±2.28

-28.47±2.80

0.1

Pim2/Hispidulin

-41.62±2.49

-15.12±3.43

23.37±3.25

-5.17±0.15

-13.35± 2.45

-25.19±2.54

-

Pim3/ GDC-0339

-42.2±2.71

-17.8±3.22

22.3±2.68

-6.71±0.16

-11.27± 2.63

-33.13±2.42

0.02

Pim3/Hispidulin

-35.15±3.28

-24.80±4.48

29.05±3.39

-5.18±0.10

-12.36± 2.36

-23.71±2.65

-

It is also a good idea to mention the limitations associated with the simulation methods, so that, other researchers will interpret the results/outcome with caution

Response: We would like to thank you to point out. We had added one sentence in the section of “2.3 Binding free energies estimated by MM/GBSA” as follow: It is noteworthy that the differences entropy contribution (TΔS) of six systems were not obvious, which implied the normal-mode analysis might be not accurate enough for Pim kinases.

Thank you very much for reconsideration.

Sincerely yours,

Jian Gao

Round 2

Reviewer 2 Report

Authors have included the required revisions as supplemental data. The manuscript looks better and may be accepted for publication.

Submitted manuscript copy shows line# 100 to 102 and 113, 114 are deleted. kindly look in to the final submission.